# The Low Sensitivity and Specificity of a Nutrition Screening Tool in Real Circumstances in a Tertiary Care Hospital Setting

**DOI:** 10.3390/children10040747

**Published:** 2023-04-19

**Authors:** Nithit Semsawat, Oraporn Dumrongwongsiri, Phanphen Phoonlapdacha

**Affiliations:** Department of Pediatrics, Faculty of Medicine Ramathibodi Hospital, Mahidol University, Bangkok 10400, Thailand; nithit.semsawat@gmail.com (N.S.); phanphen.pho@gmail.com (P.P.)

**Keywords:** nutrition screening tool, hospitalized children, STRONGkids, malnutrition, validation

## Abstract

Nutrition screening is an essential process to detect children at risk of malnutrition during hospitalization and provide appropriate nutrition management. STRONGkids is a nutrition screening tool which has been implemented in a tertiary-care hospital service in Bangkok, Thailand. This study aimed to evaluate the performance of STRONGkids in the real-situation setting. Electronic Medical Records (EMR) of hospitalized pediatric patients aged 1 month to 18 years from January to December 2019 were reviewed. Those with incomplete medical records and re-admission within 30 days were excluded. Nutrition risk scores and clinical data were collected. Anthropometric data were calculated to Z-score based on the WHO growth standard. The sensitivity (SEN) and specificity (SPE) of STRONGkids were determined against malnutrition status and clinical outcomes. In total, 3914 EMRs (2130 boys, mean age 6.22 ± 4.72 years) were reviewed. The prevalence of acute malnutrition (BMI-for-age Z-score < −2) and stunting (height-for-age Z-score < −2) were 12.9 and 20.5%. SEN and SPE of STRONGkids against acute malnutrition were 63.2 and 55.6%, stunting values were 60.6 and 56.7%, and overall malnutrition values were 59.8 and 58.6%. STRONGkids had low SEN and SPE to detect nutrition risks among hospitalized children in a tertiary-care setting. Further actions are required to improve the quality of nutrition screening in hospital services.

## 1. Introduction

The prevalence of acute malnutrition in hospitalized children ranged from 6.1 to 40.9% [1], depending on the method of data collection, study sites, or levels of healthcare. In Thailand, a nationwide prevalence of malnutrition in hospitalized children from all healthcare levels was 0.1–0.2% in 2015–2019 based on the International Classification of Diseases, 10th revision, Clinical Modification (ICD-10-CM) from the National Health Securities Office (NHSO) [2]. A higher prevalence of malnourished children was reported in more complicated healthcare settings. A previous study reported 17.3% of malnutrition in a tertiary hospital in Bangkok [3]. Previous reports of malnutrition in hospitalized children usually defined malnutrition as undernutrition or wasting, not including children with overweight and obesity.

Children who needed to be admitted to the hospital have a risk of malnutrition. They may have severe or prolonged illness, the need for sophisticated treatment, or investigation, and they may have chronic underlying diseases. There is a lot of evidence linking malnutrition in hospitalized children with adverse clinical outcomes. Malnourished children had longer hospital stays and higher hospital costs than children with normal nutrition status [4,5]. Malnutrition also leads to higher infection and mortality rates among hospitalized children [6,7]. Both innate and adaptive immunity are impaired by malnutrition and nutrient deficiencies. Consequently, immune dysfunction contributed to infection, mortality, and morbidity among malnourished children [8].

In order to provide appropriate nutritional support during hospitalization, it is important to assess nutrition risk at admission to hospital. Currently, many pediatric nutrition screening tools have been developed and integrated in routine healthcare services, for example the Pediatric Subjective Global Nutrition Assessment (SGNA) [9], and the Screening Tool for the Assessment of Malnutrition in Pediatrics (STAMP) [10], the Screening Tool for Risk on Nutritional Status and Growth (STRONGkids) [11]. Validation studies have been carried out to determine the performance of the screening tools; however, there is no consensus for the most appropriate method of assessing nutrition risk.

STRONGkids is a nutrition screening tool developed by Hulst et al. in the Netherlands in 2010 [11]. It is a practical, easy, and reliable tool for the assessment of nutrition risk [12], with high sensitivity and specificity against malnutrition status [3,13]. In addition to malnutrition status, a previous study showed the association between the nutrition risk score assessed by STRONGkids with clinical outcomes such as hospital stay and cost [14]. STRONGkids is a questionnaire consisting of four yes-no questions including (1) subjective clinical assessment, (2) high risk disease, (3) nutrition intake and loss, and (4) weight loss or poor weight gain. Each questions scores 1 point, except for high-risk disease, which scores 2 points. The nutrition risk scores are ranged from 0 to 5. Then, nutrition risk scores are translated to nutrition risk classification as low risk (score = 0), moderate risk (score = 1–3), and high risk (score = 4–5) [11]. Nutrition risk classifications guide the allocation of appropriate nutrition management, especially in large hospitals. This questionnaire was adopted and translated into the Thai language and validated by the Faculty of Medicine Siriraj Hospital, Mahidol University (manuscript in preparation). Since 2019, the Thai version of STRONGkids has been implemented in routine healthcare service at Ramathibodi Hospital, Mahidol University.

Ramathibodi Hospital is a 1000-bed university hospital located in Bangkok, providing sophisticated healthcare services. Most patients suffered from complicated medical conditions and underlying chronic diseases. Since the nutrition screening tool was developed and validated in primary and secondary healthcare settings [11], the performance of the screening tool in complicated tertiary care settings has been questionable. Moreover, the ability to detect malnutrition may differ between the validation study and the real circumstances. This study aimed to validate STRONGkids as a nutrition screening tool in a highly sophisticated tertiary hospital while the tool was implemented in the hospital service system. We also aimed to investigate the relationship between nutrition risks and clinical outcomes.

## 2. Materials and Method

### 2.1. Study Designs, Sites and Participant Characteristics

This was a retrospective study. Electronic medical records (EMR) of children admitted to the pediatric ward at Ramathibodi Hospital, Bangkok, Thailand, from 1 January to 31 December 2019 were reviewed. EMRs of patients aged under 30 days or over 18 years, lack of nutrition screening records and anthropometric data, and re-admission within 30 days were excluded. Infants aged under 30 days were admitted to the sick-newborn ward, which had a different nutrition risk assessment from the pediatric ward. Adolescents aged over 18 years might be admitted to either the pediatric or adult ward, but the growth assessments (detail in anthropometric data below) were different from children aged under 18. Re-admission within 30 days was excluded to prevent data repetition. We expected that the clinical and nutritional status of patients would remain similar during a period of 30 days. Since the study aimed to explore the nutrition risk scores of all admissions occurred through the year 2019, sample size calculation was not performed. All data were gathered and secured in the Research Electronic Data Capture (REDCap^®^) platform. Patient characteristics including age, gender, underlying disease(s), and clinical data were collected from the EMR. In routine hospital service, admissions are categorized into 2 groups according to reason for admission as either admission due to acute illness or scheduled admission for medical intervention.

### 2.2. The Nutrition Risk Score and Anthropometric Data

The nutrition risk score and classifications (low, moderate, high risk) were obtained from the STRONGkids record which was routinely filled by a responsible nurse on admission. STRONGkids records were kept along with all admission records in the EMRs.

Anthropometric assessments at admission, including weight and height/length measurements, were routinely performed by nurses at all pediatric wards with standard techniques. The equipment used for weight measurement was routinely calibrated by the hospital equipment services. Weight and height/length data were recorded in the admission document. We collected the anthropometric data at admission from EMRs. Patients’ weight and height/length were calculated to body mass index (BMI), BMI-for-age Z-score (BAZ) and height/length-for-age Z-score (HAZ) according to the WHO growth standard by the program WHOAnthro (for children aged 0–5 years) and WHOAnthroPlus (for children aged 5–18 years).

Children with BAZ under −2 were classified as acute malnutrition, and those with HAZ under −2 were classified as stunting. As using the weight-for-height Z-score to determine malnutrition is limited to age 0–5 years, we used BAZ to broaden the diagnosis for all study populations.

### 2.3. Clinical Outcomes

Clinical outcomes, including length of hospital stay (LOS), mortality, hospital-acquired infection, and hospital cost, were reviewed from EMRs. LOS was collected from the discharge summary of each admission in days and hours. Mortality was counted from the death event that occurred in admission, and did not include patients discharged with end of life care at home. Hospital-acquired infection included hospital-acquired pneumonia (HAP), catheter-associated blood stream infection (CABSI), and urinary tract infection (UTI), which were recorded in the EMRs and had proved culture positive. Hospital cost (in Thai Baht; THB) was the total spending for treatment in each admission.

### 2.4. Statistical Analysis

Prior to analysis, all data were exported from the REDCap^®^ platform to an excel file. Then, data were explored, checked for incorrect and missing data, and were corrected. Data analysis was accomplished in STATA version 17.0. Descriptive data were presented as mean and median according to data distribution. The comparison of clinical outcomes by nutrition risk classification was carried out by the Pearson’s chi-square test or Fisher’s exact test for categorical data (mortality and hospital-acquired infection) and the Kruskal–Wallis test for continuous data (hospital stay and cost). A *p*-value < 0.05 was considered statistical significance. Sensitivity (SEN), specificity (SPE), positive predictive value (PPV), and negative predictive value (NPV) of STRONGkids were analyzed. For the validity test, the nutrition risk scores were classified into 2 categories as low risk (score 0) and medium-to-high risk (score 1–5). Since there was no consensus of a gold standard parameter for the determination of malnutrition in hospitalized patients, we used many parameters to determine the validity of the screening test. The concurrent validity of STRONGkids was determined against the nutritional status of patients, including acute malnutrition, stunting, and overall malnutrition (patients who had either acute malnutrition or stunting). We also analyzed the predictive validity of STRONGkids against clinical outcomes including mortality, hospital-acquired infection, and LOS. The receiver operating characteristic (ROC) curve was analyzed to determine the performance of the screening test against malnutrition status.

## 3. Results

From 5529 admissions in 2019, 877 admissions (15.86%) were excluded according to the exclusion criteria, as shown in Figure 1. After the EMRs were reviewed, 738 records (15.86%) were excluded due to a lack of STRONGkids records and incomplete/inaccurate anthropometric records. Finally, data were collected from 3914 EMRs. Baseline characteristics and anthropometric data are presented in Table 1. The mean age of children was 6 years old. Approximately 60% of admissions were scheduled for investigations, surgical interventions, and the administration of medications or chemotherapy. Three quarters of children had underlying chronic diseases. The prevalence of acute malnutrition and stunting were 12.9 and 20.5%, respectively. Overall malnutrition was found in 29.7% of hospitalized children during the study period.

### Nutrition Risk Classification

According to nutrition screening by STRONGkids, low, moderate, and high nutrition risk classifications were found in 2081 (53.17%), 1727 (44.12%), and 106 (2.71%) admission records, respectively. When classified by reason of admission, age group, and underlying diseases, the proportion of moderate and high risk was higher among children admitted with acute illness compared to those admitted with scheduled admission, aged below 5 years compared to age 5–18 years, and in children with underlying chronic diseases compared to children with no underlying diseases, as shown in Figure 2.

The comparisons of LOS, hospital cost, mortality, and hospital-acquired infection among admissions with different nutrition risks are shown in Table 2. High nutrition risk was associated with higher LOS, hospital cost, mortality, and hospital-acquired infection.

#### Validation of Nutrition Screening Tool

The concurrent and predictive validity values of STRONGkids are shown in Table 3. The SEN and SPE of the screening tool against nutritional status, including acute malnutrition, stunting, and overall malnutrition, were approximately 55–63%. The analysis of predictive validity shower higher SEN against mortality (86.2%) and hospital-acquired infection (78.6%). The ROC curves of STRONGkids are shown in Figure 3. The area under the curves were 0.594, 0.587, and 0.592 when analysis was performed against acute malnutrition, stunting, and overall malnutrition, respectively.

There was a high proportion of children with scheduled admission, which might affect the nutrition risk assessment. The validity of STRONGkids was analyzed against malnutrition status among children admitted to the hospital with acute illness, as shown in Table 4. SEN against acute malnutrition, stunting, and overall malnutrition was higher than the analysis of overall participants.

## 4. Discussion

This present study showed that the performance of the nutrition screening tool in our real hospital circumstance was not satisfactory, as we found the SEN and SPE of STRONGkids against malnutrition status to be 59.8–63.2%, and 55.6–58.6%, respectively. In addition, our findings showed that the prevalence of overall malnutrition, acute malnutrition, and stunting were 29.7%, 12.9%, and 20.5%, respectively. 

The prevalence of malnutrition in hospitalized patients from our study was higher than those previously reported [1]. The prevalence of malnutrition was also higher than that reported by NHSO [2]. The national data were based on ICD-10 cm codes entered into the computer system, which may be missing or malnutrition may not have been realized by the responsible physicians. Our setting is a tertiary hospital, which cares for children with complicated diseases. Almost three quarters of hospital admissions were children with chronic underlying diseases which affect their nutritional status, both in the short and long term. Therefore, a high prevalence of both acute malnutrition and stunting was found in our study. Tertiary-care hospitals in Thailand reported a higher rate of malnutrition than NHSO data [2]. 

Previous studies regarding the validation of STRONGkids showed good SEN against malnutrition while using similar nutrition status indicators as our study [3,15,16]. Santos et al. [16] found that SEN was 94.1% and SPE was 16.3%, while Ortiz-Gutierrez et al. [15] found the SEN was 86% and SPE was 72%. A study from Thailand showed high SEN (95.7%) but low SPE (22.8%) [3]. According to these previous findings, the performance of STRONGkids as a nutrition screening tool for hospitalized children was not questionable. However, these validation studies were prospective studies, and a group of researchers evaluated the nutrition risk score themselves. In contrast, our study collected the data from the real-situation setting. There were many nurses who were responsible for the assessment of the nutrition risk scores, which might not have been standardized. In addition, there were other factors which might affect the quality of nutrition screening evaluation, such as workload and an understanding of how to perform nutrition screening by nurses. STRONGkids items no. 1 (subjective clinical assessment) and no. 2 (high risk diseases) may be problematic. Many nurses were hesitant to determine yes or no on the subjective clinical assessment when a child looked wasting but had edema or ascites. We found that among children with BAZ under −2, only 22.4% had a score = 1 in the subjective clinical assessment. In addition, 2.7% of children with BAZ more than 2 had a score = 1 on this item. The nurses may also have been confused by the list of high-risk underlying diseases, and might not have had enough time to review the medical history of children. Therefore, they might provide inaccurate assessments on this item. Among children with underlying chronic illness, 37.2% had a score = 2 in the high-risk diseases item in the nutrition risk assessment. To improve the quality of the nutrition risk assessment, we need to ensure correct understanding of the nutrition screening tool among responsible nurses or standardize a group of personnel to perform nutrition risk screening in routine hospital services.

Another factor which was associated with the performance of screening tools to detect malnutrition in our setting was patient characteristics; this may differ from previous studies. There were limitations to performing an accurate anthropometric assessment in some children with chronic and complicated diseases. Weight measurements in children with organomegaly, mark ascites, or limb loss did not reflect their actual nutrition status. Height measurements in children who were bedridden, had joint contracture or scoliosis provided inaccurate results. There were some patients who suffered from chromosomal anomalies or endocrine diseases, which might affect their height. 

Patient status at admission might affect nutrition risk assessment. Children who were admitted for scheduled intervention or therapeutic purposes had better clinical status than children admitted with acute illness. Meanwhile, they usually suffered from chronic underlying diseases. We found that the proportion of nutrition risk classification was different between scheduled admission and admission due to acute illness (Figure 2). As 60% of admissions in this study were scheduled admissions, this may also affect the performance of the nutrition screening tool. When scheduled admissions were excluded, the analysis of the screening tool validity showed better SEN (Table 4). 

Our study showed that a higher nutrition risk score was associated with worse clinical outcomes. LOS, mortality, and infection rates were higher with an increased nutrition risk score. Similarly, a study of nutrition risk screening in a tertiary care center showed an increase in hospital stay and cost among patients with high nutrition risk [14]. Diamanti et al. [4] found that malnutrition was significantly associated with an 86% increase in risk of prolonged hospitalization. These findings emphasized the need for appropriate nutritional management among patients with high nutrition risk to decrease complications and mortality. The results of this study were obtained from admissions during 2019, when the nutrition screening tool was first implemented in the hospital. In that period, the nutrition supporting system according to the nutrition risk score was not well-established. Therefore, we decided to collect the clinical outcomes, including mortality, infection rate and hospital stay, and analyzed predictive validity against these factors. Effective nutrition intervention, especially in high nutrition risk children, may modify their clinical outcomes during hospitalization.

This is the first study regarding the validation study of the Thai version of STRONGkids as a nutritional screening tool in a real hospital circumstance with a large number of admission records. We demonstrated some barriers and difficulties in using this screening tool in routine hospital services, which leads to quality improvements to be made in the future. Nevertheless, there were some limitations in our study. Firstly, STRONGkids and anthropometric assessment records were absent in approximately 13% of EMRs, which we then needed to exclude from the study. Secondly, we used nutrition status at admission as an indicator for the validation of the screening tool, while there might be more factors associated with malnutrition during hospitalization. To determine the risk of malnutrition in hospitals, dynamic nutrition assessment records should be obtained as the gold standard for validation study. Unfortunately, we lacked longitudinal nutrition assessment data from retrospective medical record reviews.

Although STRONGkids showed good SEN in previous organized validation studies, our study showed that the performance of this nutrition screening tool in our hospital setting was unsatisfactory. This might be due to the competency of the healthcare personnel who performed the nutrition screening, or the nutrition screening tools themselves, which may not be suitable for highly sophisticated hospital services. In order to improve the quality of nutrition screening in hospital services, the standardization of healthcare personnel who are responsible for nutrition screening in pediatric wards should be carried out and the screening tool’s performance re-evaluated.

## 5. Conclusions

High nutrition risk scores showed associations with adverse clinical outcomes. From our validation study in the real tertiary-care hospital circumstance, STRONGkids had low SEN and SPE to detect malnutrition among hospitalized pediatric patients. These findings may be due to the nutrition screening performance of the responsible healthcare personnel, or to the clinically different characteristics of patients admitted to the hospital. Further action and study are required to improve the quality of nutrition screening in hospital services.

## Figures and Tables

**Figure 1 children-10-00747-f001:**
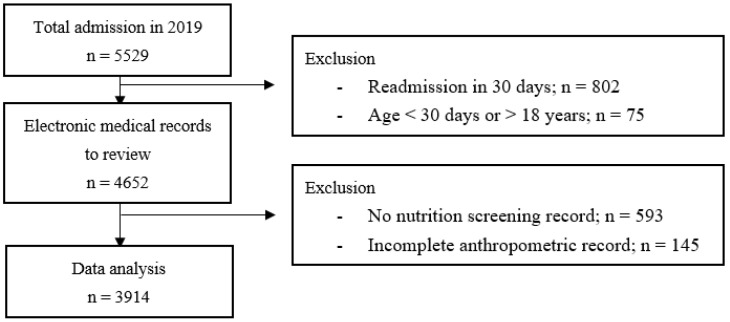
Study enrollment of electronic medical records.

**Figure 2 children-10-00747-f002:**
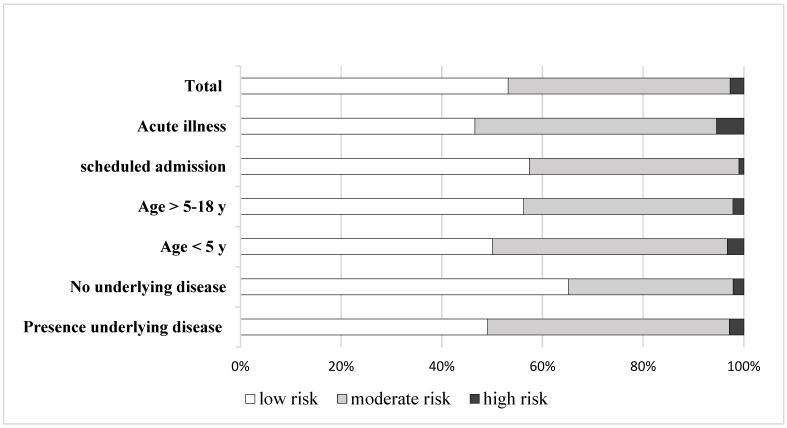
Nutrition risk score (STRONGkids) classified by patients’ characteristics.

**Figure 3 children-10-00747-f003:**
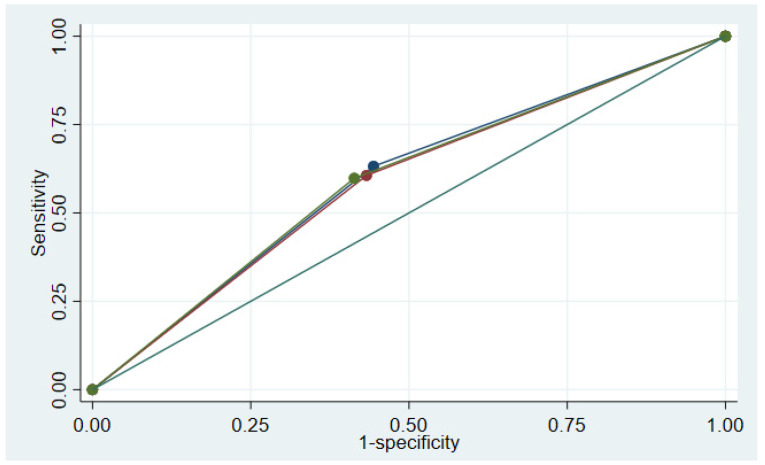
Receiver operation characteristic (ROC) curve of STRONGkids against acute malnutrition (green), stunting (blue), and overall malnutrition (red).

**Table 1 children-10-00747-t001:** Baseline characteristics and anthropometric data of 3914 electronic medical records.

Characteristics	Results
Mean age (y); mean (SD)	6.22 (4.72)
Gender; *n* (%)	
Male	2130 (54.42)
Female	1784 (45.58)
Reason for admission; *n* (%)	
Scheduled admission	2387 (60.99)
Acute illness	1527 (39.01)
Presence of chronic underlying disease; *n* (%)	2929 (74.83)
Anthropometric parameters; mean (SD)	
Weight-for-age Z-score (age < 10 y; *n* = 2979)	−0.65 (1.68)
Weight-for-height Z-score (age < 5 y; *n* = 1957)	−0.26 (1.74)
Height-for-age Z-score	−0.91 (1.53)
BMI-for-age Z-score	−0.13 (1.82)
Anthropometric assessment; *n* (%)	
Acute malnutrition	505 (12.90)
Stunting	802 (20.49)
Overall malnutrition	1162 (29.70)

**Table 2 children-10-00747-t002:** Comparisons of clinical outcomes by nutrition risk classification.

Clinical Outcomes	Nutrition Risk ^1^			*p*-Value
	Low Risk (N = 2081)	Moderate Risk (N = 1727)	High Risk (N = 106)	
Hospital stay (days) Median (IQR)	2.2 (1.7–4.0)	2.9 (1.9–6.0)	7.1 (3.8–18.6)	<0.001
Hospital cost (THB) Median (IQR)	18,258 (10,873–28,154)	23,441 (11,743–52,194)	42,750 (19,897–138,885)	<0.001
Mortality; *n* (%)	4 (0.19)	21 (1.22)	4 (3.77)	<0.001
Hospital-acquired Infection; *n* (%)	15 (0.72)	47 (2.72)	8 (7.55)	<0.001
HAP	6 (0.29)	22 (1.27)	5 (4.72)	<0.001
CABSI	3 (0.14)	9 (0.52)	3 (2.83)	<0.001
UTI	6 (0.29)	18 (1.04)	3 (2.83)	0.001

^1^ Classification of nutrition risk scores screened by STRONGkids: low risk—score 0; moderate risk—score 1–3; high risk—score 4–5. HAP; hospital acquired pneumonia, CABSI; catheter-related blood stream infection, UTI; urinary tract infection, THB; Thai baht (USD 0.03).

**Table 3 children-10-00747-t003:** Concurrent and predictive validity of STRONGkids.

	SEN [95% CI]	SPE [95% CI]	PPV [95% CI]	NPV [95% CI]
**Concurrent validity**				
Acute malnutrition	63.2 [58.8–67.4]	55.6 [53.9–57.3]	17.5 [15.8–19.3]	91 [89.7–92.2]
Stunting	60.6 [57.1–64]	56.7 [55–58.5]	26.5 [24.5–28.6]	84.8 [83.2–86.3]
Overall malnutrition	59.8 [56.9–62.6]	58.6 [56.8–60.5]	37.9 [35.7–40.2]	77.6 [75.7–79.3]
**Predictive validity**				
Mortality	86.2 [68.3–96.1]	53.5 [51.9–55]	1.36 [0.89–2.01]	99.8 [99.5–99.9]
Hospital-acquired infection rate	78.6 [67.1–87.5]	53.7 [52.2–55.3]	3 [2.27–3.89]	99.3 [98.8–99.6]
Longer hospital stay ^1^	55.2 [53–57.5]	61.5 [59.3–63.6]	58.6 [56.4–60.9]	58.1 [56–60.3]
Hospital stay over 14 days	69.8 [64.1–75]	55 [53.4–56.6]	11.1 [9.67–12.6]	95.8 [94.8–96.6]

^1^ Hospital stay longer than median (2.3 days). SEN; sensitivity, SPE; specificity, PPV; positive predictive value, NPV; negative predictive value.

**Table 4 children-10-00747-t004:** Concurrent validity of STRONGkids among children admitted due to acute illness.

	SEN [95% CI]	SPE [95% CI]	PPV [95% CI]	NPV [95% CI]
Acute malnutrition	71.5 [65.1–77.3]	49.6 [46.9–52.4]	19.4 [16.7–22.3]	91.1 [88.8–93.1]
Stunting	68.0 [62.1–73.5]	49.8 [47.0–52.6]	22.9 [20.1–26.0]	87.6 [85.0–90.0]
Overall malnutrition	67.4 [62.7–71.8]	52.1 [49.0–55.1]	35.7 [32.4–39.1]	80.2 [77.0–83.0]

SEN; sensitivity, SPE; specificity, PPV; positive predictive value, NPV; negative predictive value.

## Data Availability

The analyzed dataset is available upon request from the corresponding author.

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
