# Peer review of "The Low Sensitivity and Specificity of a Nutrition Screening Tool in Real Circumstances in a Tertiary Care Hospital Setting"

_children, 2023, doi:10.3390/children10040747_

Round 1

Reviewer 1 Report

Abstract:

The authors mentioned that "STRONGkids is a nutrition screening tool which has been implemented in our tertiary-care hospital service". I would suggest mentioning the name of the country instead of writing "our" tertiary-care hospital services.

Introduction:

What are the reasons for variations of existing national evidence on the prevalence of malnutrition (0.1-0.2% in 2015-2019) with other studies (17.1%). 

I would suggest defining malnutrition first in the introduction. Malnutrition carries dual meaning i.e. undernutrition and overweight. It is important to define first and then explain the prevalence etc., 

What is the definition of hospitalized children? I would also suggest mentioning the definition in the introduction section. 

Currently, many pediatric nutrition screening tools were developed and integrated into routine healthcare services. Mention the references and also mention the name of pediatric nutrition screening tools as well.  

2nd last paragraph: The last two sentences of this para should copy and discuss in the method section. I would suggest discussing more measuring scales and their importance in the introduction section. 

Methods:

What is the study design to carry out this study? This information is missing?

Study site: Add the headline and clearly write about the study design. 

Data source: Clearly mention the data source and how did the authors access this data source. Is it publically available data? 

study variables: details are missing. Mention all study variables

Recording the variables: The detail is not much clear. 

Data cleaning: is not mentioned

Sample size: not mentioned in the method's section

Discussion

In the discussion section, the authors are claiming that the screening tool in our real hospital circumstance was not satisfied. However, the authors did not provide/discuss the justification for such results.

The author did not discuss the inclusion and exclusion criteria that may be the reasons for not optimal results by using this tool. I would also suggest discussing this in the methods section. 

Study strengths and limitations must be clearly spelled out in the discussion.

Conclusions:

As a researcher suggests practical recommendations instead of just writing further action and study are required to improve the quality of nutrition screening in hospital services.

Author Response

Thank you for your suggestion. I would like to response to your comment. Please find it in the attached file.  

Thank you for your suggestion. I would like to response to your comment as follows:

Abstract:

The authors mentioned that "STRONGkids is a nutrition screening tool which has been implemented in our tertiary-care hospital service". I would suggest mentioning the name of the country instead of writing "our" tertiary-care hospital services.

I revised the abstract according to your suggestion

Introduction:

What are the reasons for variations of existing national evidence on the prevalence of malnutrition (0.1-0.2% in 2015-2019) with other studies (17.1%).

The prevalence from the National database was low because the study collected the data from ICD10 record in the National Health database. Physicians sometimes do not enter the ICD code for malnutrition in the record, especially in the primary care hospital. I also discussed this issue in the discussion part.

I would suggest defining malnutrition first in the introduction. Malnutrition carries dual meaning i.e. undernutrition and overweight. It is important to define first and then explain the prevalence etc.,

I agree with you that malnutrition both under- and over-nutrition. I added the explanation in the manuscript.

What is the definition of hospitalized children? I would also suggest mentioning the definition in the introduction section.

I explained more regarding the hospitalized children in the introduction and method.

Currently, many pediatric nutrition screening tools were developed and integrated into routine healthcare services. Mention the references and also mention the name of pediatric nutrition screening tools as well. 

The information regarding pediatric nutrition screening tool was added in the manuscript.

2nd last paragraph: The last two sentences of this para should copy and discuss in the method section. I would suggest discussing more measuring scales and their importance in the introduction section.

I added the detail of the measuring scale and the importance of nutrition risk screening in the introduction.

Methods:

What is the study design to carry out this study? This information is missing?

This is a retrospective study. The information was added in the manuscript.

Study site: Add the headline and clearly write about the study design.

The headline was added to the manuscript.

Data source: Clearly mention the data source and how did the authors access this data source. Is it publically available data?

Data was reviewed from Electronic Medical Records (EMRs) which is a hospital database of all patients receiving care from the hospital. It is publically available because of patient confidentiality issue. Person who can assess the data is one who responsible for caring patients in the hospital or is authorized to access for research or academic purposes. 

Study variables: details are missing. Mention all study variables

Recording the variables: The detail is not much clear.

The study variables were participant characteristics, anthropometric data, nutrition risk score, and clinical outcomes. They were reviewed from the admission record in the EMRs. The detail and definition were mentioned in the method section.

Data cleaning: is not mentioned

I added data cleaning in the manuscript.

Sample size: not mentioned in the method's section

Sample size estimation was not performed. Since we would like to explore the nutrition risk score of all admission records in the year 2019.

Discussion

In the discussion section, the authors are claiming that the screening tool in our real hospital circumstance was not satisfied. However, the authors did not provide/discuss the justification for such results.

Low sensitivity and specificity of the screening tool from our findings was probably due to unstandardized assessors and inaccurate anthropometric measurements among patients with complicated clinical presentation, which was already discussed.

The author did not discuss the inclusion and exclusion criteria that may be the reasons for not optimal results by using this tool. I would also suggest discussing this in the methods section.

The discussion was added in the method section.

Study strengths and limitations must be clearly spelled out in the discussion.

Strengths and limitation were added in the discussion

Conclusions:

As a researcher suggests practical recommendations instead of just writing further action and study are required to improve the quality of nutrition screening in hospital services.

Practical recommendation were added in the discussion and conclusion

Reviewer 2 Report

It is an interesting study on a common problem, malnutrition in Hospitals, particularly those of tertiary care. Some observations:

Results

When authors mention that approximately 60% of admissions were scheduled for investigations, do you mean that they were admitted for diagnostic procedures or research protocols?

According to your results, almost 75% had an underlying chronic disease. However, 53% have a low risk rating (low risk = 0 points), when having an underlying disease with a risk of malnutrition is equivalent to 2 points. Please clarify.

Figure 2: The title should say: Nutrition risk score (STRONGkids) classified by patient´s characteristics.

Table 2. Add the meaning of the abbreviation THB (Hospital cost).

Discussion

Authors should add to the discussion the fact that these results have internal validity, particularly for the Hospital in which it was performed; with the limitations of the evaluation that was carried out.

About the first question of the STRONGkids questionnaire (¿Is the patient in a poor nutritional status judged by subjective clinical assessment?), discuss about specific alterations of chronic pathologies, which may present with ascites, visceromegaly, hydrocephalus, or special conditions such as cerebral palsy, in whom the evaluation based on subjective clinical assessment could be difficult to interpret.

Author Response

Thank you for your suggestion. I would like to response to your comment. Please find it in the attached file.  

Thank you for your suggestion. I would like to response to your comment as follows:

Results

When authors mention that approximately 60% of admissions were scheduled for investigations, do you mean that they were admitted for diagnostic procedures or research protocols?

Scheduled admission meant that patient was admitted for medical intervention or medication in routine hospital services. This study was retrospective study, there was no intervention in research protocol. I added more explanation regarding study design and detail of scheduled admission in the manuscript.

According to your results, almost 75% had an underlying chronic disease. However, 53% have a low risk rating (low risk = 0 points), when having an underlying disease with a risk of malnutrition is equivalent to 2 points. Please clarify.

The percentage of children with underlying chronic diseases and children with risk score = 2 in the item of having underlying diseases were different due to

1) According to STRONGkids, not all underlying chronic diseases are considered to be a risk of malnutrition. Some children with chronic diseases had score = 0 in this item if their diseases were not in the list.

2) Underlying chronic diseases were reviewed from EMRs by the researchers. While risk score were performed at admission by responsible nurses. This was also the problem mentioned in the discussion that there might be some mistakes in risk assessment by healthcare personnel in the real circumstance, which needed to be standardized. 

Figure 2: The title should say: Nutrition risk score (STRONGkids) classified by patient´s characteristics.

The title of figure 2 was corrected according to your suggestion

Table 2. Add the meaning of the abbreviation THB (Hospital cost).

The meaning of the abbreviation THB was added according to your suggestion

Discussion

Authors should add to the discussion the fact that these results have internal validity, particularly for the Hospital in which it was performed; with the limitations of the evaluation that was carried out.

The discussion regarding the limitation of risk assessment by nurses were added in the discussion section.

About the first question of the STRONGkids questionnaire (¿Is the patient in a poor nutritional status judged by subjective clinical assessment?), discuss about specific alterations of chronic pathologies, which may present with ascites, visceromegaly, hydrocephalus, or special conditions such as cerebral palsy, in whom the evaluation based on subjective clinical assessment could be difficult to interpret.

The discussion was added in the manuscript.

Round 2

Reviewer 1 Report

Data availability

Please check the below statement. Were the data publicly available?  Moreover please correct the spelling of publicly. 

"It is publically available because of patient confidentiality issue".

Mention the sample size/observation of admission records in the year 2019. Mentioning the sample size is important to know the power of the findings. 

Author Response

Thank you for your suggestion. Please find the response to your comment in the attached file. 
